# Stabilizing the closed SARS-CoV-2 spike trimer

Jarek Juraszek [1,3], Lucy Rutten [1,3], Sven Blokland [1], Pascale Bouchier [1], Richard Voorzaat [1], Tina Ritschel [1], Mark J. G. Bakkers [1], Ludovic L. R. Renault [2] & Johannes P. M. Langedijk [1✉]

The trimeric spike (S) protein of SARS-CoV-2 is the primary focus of most vaccine design and development efforts. Due to intrinsic instability typical of class I fusion proteins, S tends to prematurely refold to the post-fusion conformation, compromising immunogenic properties and prefusion trimer yields. To support ongoing vaccine development efforts, we report the structure-based design of soluble S trimers with increased yields and stabilities, based on introduction of single point mutations and disulfide-bridges. We identify regions critical for stability: the heptad repeat region 1, the SD1 domain and position 614 in SD2. We combine a minimal selection of mostly interprotomeric mutations to create a stable S-closed variant with a 6.4-fold higher expression than the parental construct while no longer containing a heterologous trimerization domain. The cryo-EM structure reveals a correctly folded, predominantly closed pre-fusion conformation. Highly stable and well producing S protein and the increased understanding of S protein structure will support vaccine development and serological diagnostics.

[1] Janssen Vaccines & Prevention B.V., Archimedesweg 4-6, Leiden, The Netherlands. [2] NeCEN, Leiden University, Einsteinweg 55, Leiden, The Netherlands. [3] These authors contributed equally: Jarek Juraszek, Lucy Rutten. ✉email: hlangedi@its.jnj.com

Development of effective preventative interventions against the SARS-CoV-2 virus that causes the ongoing COVID-19 pandemic[1,2] is urgently needed. The viral surface spike (S) protein is a key target for prophylactic measures as it is critical in the viral life cycle and the primary target of neutralizing antibodies[3–6]. S is a large, trimeric glycoprotein that mediates both binding to host cell receptors and fusion of virus and host cell membranes, through its S1 and S2 subunits, respectively[7–9]. The S1 subunit comprises two distinct domains: an N-terminal domain (NTD) and a host cell receptor-binding domain (RBD). For infection, S requires proteolytic cleavage by a furin-like protease between the S1 and S2 subunits (S1/S2), and by TMPRSS2 at a conserved site directly preceding the fusion peptide (S2′)[10,11]. In the prefusion state, the S-protein's RBD domains alternate between open ('up') and closed ('down') conformations. The receptor-binding site, which can bind to human angiotensin-converting enzyme 2 (ACE2), is transiently exposed in the 'up' conformation. Like other class I fusion proteins, the SARS-CoV-2 S protein is intrinsically metastable as a consequence of its ability to undergo extensive conformational changes that are required to drive fusion.

The prefusion conformation of S as present on the infectious particle contains the epitopes for neutralizing antibodies and thus holds most promise as a vaccine immunogen[3–6,12,13]. Prefusion stabilization typically increases the recombinant expression of viral glycoproteins, which facilitates the production of protein for (subunit) vaccines and improves the immune response elicited by recombinant protein and viral vector vaccines[14]. In recent years, efforts have been made to stabilize various class I fusion proteins through structure-based design (for a review see ref. [14]). A particularly successful approach to enhance prefusion stability was shown to be the stabilization of the so-called hinge loop preceding the central helix (CH), which has been applied to a range of viral fusion glycoproteins[15–20]. Stabilization of the hinge loop of the S proteins of SARS-CoV and MERS-CoV has been achieved by mutation of two consecutive residues in the S2 subunit between the central helix (CH) and heptad repeat 1 (HR1)[21,22] to proline and this approach (2P) has successfully been applied to the SARS-CoV-2 S protein[9]. However, the SARS-CoV-2 S protein carrying these substitutions and additional furin site mutations (S-2P) remains unstable and strategies to further improve its stability have recently been described[12,23–25]. Comparison of the structure of SARS-CoV-2 S-2P[9,26] with that of native SARS-CoV-2 S[27,28] shows that the former adopts a more open conformation with one or more of the RBDs in the 'up' conformation. Although neutralizing antibodies have been mapped to the RBD in the up as well as down state, the antibodies that bind the conserved epitopes on the RBD in the down state were described to have the highest neutralizing potency[4,6,13]. Therefore, stabilizing S in its closed (3 RBDs down) state and arresting the first step in the conformational change may result in an improved vaccine immunogen.

In this work, using structure-based design, we find stabilizing mutations in both the S1 and S2 subdomains. Combining several of the mutations results in a highly stable S trimer, S-closed, with increased expression that remains stable in the absence of a heterologous trimerization domain that is typically required in soluble S designs[9,21,26,29]. Assessment of its antigenicity and high-resolution EM confirm that this trimer adopts a closed conformation.

## Results

To stabilize the SARS-CoV-2 S protein in the closed pre-fusion state, we took a rational approach based on the structure of S[9]. We searched for cavity-filling substitutions, buried charges, and

possibilities for forming disulfides. We expanded our search for proline and glycine mutations, beyond the previously described hinge loop in the S-2P variant. Mutations were identified computationally with Rosetta's mutant design[30] and Bioluminate cysteine bridge scanning[31] followed by visual inspection of molecular interactions. Selected mutations are presented in Supplementary Fig. 1 and fall in two structural categories-SD1 head mutations N532P, T572I, D614N, and S2 loop mutations A942P, T941G, T941P, S943G, A944P, A944G (loop α13α14), and A892P (loop α10α11). Disulfides F888C + G880C (DS1) and S884C + A893C (DS2) were identified in loop α10α11.

S ectodomain variants with mutations according to Supplementary Fig. 1 were expressed as single chain by mutation of the furin site and addition of a C-terminal foldon (Fd) trimerization domain without (ΔFurin) or with the two previously described stabilizing prolines in the hinge loop (S-2P)[9]. Supernatants of Expi293F cells transfected with plasmids encoding the S variants were tested for trimer content (Fig. 1a) and for RBD exposure of the S protein by ACE2 binding (Fig. 1b). All mutations significantly increased trimer yields and ACE2 binding of the S protein for both the ΔFurin and S-2P variants. A strong effect was observed with T941P, A942P, and A944P. A942P showed an ~11-fold increase in expression for ΔFurin, and ~3-fold for S-2P (Fig. 1a). For T941P, A944G, and K986P shorter retention times were observed, likely indicating an opening of the trimers. T941P and A944G showed the highest ACE2 binding among the α13α14 loop mutations and K986P resulted in ~10-fold higher ACE2 binding, whereas the trimer yield was only ~3-fold higher than ΔFurin. This indicates that the RBD domains are more exposed.

The stability of the single point mutants was further characterized by purified proteins using differential scanning calorimetry (DSC). First, we tested the contribution to the stability of the individual proline mutations of the S-2P variant (Fig. 1c left panel, Supplementary Table 1). All curves showed two melting events ($Tm_{50}$'s), albeit with different ratios. ΔFurin and additional V987P show a major $Tm_{50}$ at 64 °C and a minor $Tm_{50}$ around 49 °C. This is inverted for the K986P mutant in which the lower temperature transition is dominant. The combination of both prolines reduced the lower and increased the higher $Tm_{50}$. Among the S1 mutants (Fig. 1c central panel and Supplementary Table 1) D614N and the natural variant D614G diminished the peak height of the first $Tm_{50}$ and increased the second by almost 2 °C in their respective backgrounds, which was subsequently confirmed with differential scanning fluorimetry (DSF) for D614N (Supplementary Fig. 2). A similar effect was observed for T572I and the loop-stabilizing mutations A892P and DS1, albeit to a lesser extent (Fig. 1c right panel and Supplementary Table 1). A942P, which increased the trimer yields, hardly affected the thermal stability in DSC (Fig. 1c right panel) or DSF (Supplementary Fig. 2).

RBD exposure was characterized by binding of ACE2, neutralizing antibody SAD-S35 and non-neutralizing antibody CR3022 that competes with ACE2. ACE2 and SAD-S35 can only bind RBD in the 'up' configuration and CR3022 can only bind when 2 RBDs are in the 'up' configuration[5]. The variant with K986P showed higher binding of SAD-S35, ACE2, and CR3022 than S-2P, measured with BioLayer Interferometry (Fig. 1d left panel), in accordance with the results obtained with SEC and AlphaLISA, both in cleared crude supernatants. D614N and T572I showed very low binding to SAD-S35 and ACE2 and almost no CR3022 binding (Fig. 1d right panel), indicating more closed trimers. A892P improved trimer closure to a lesser extent, while A942P seemed to increase its opening. These mutants likely exhibit a mixture of closed, 1-up, and 2-up structures.

The purified S variants with higher $Tm_{50}$ and lower Ab binding compared to S-2P also showed longer retention times in SEC

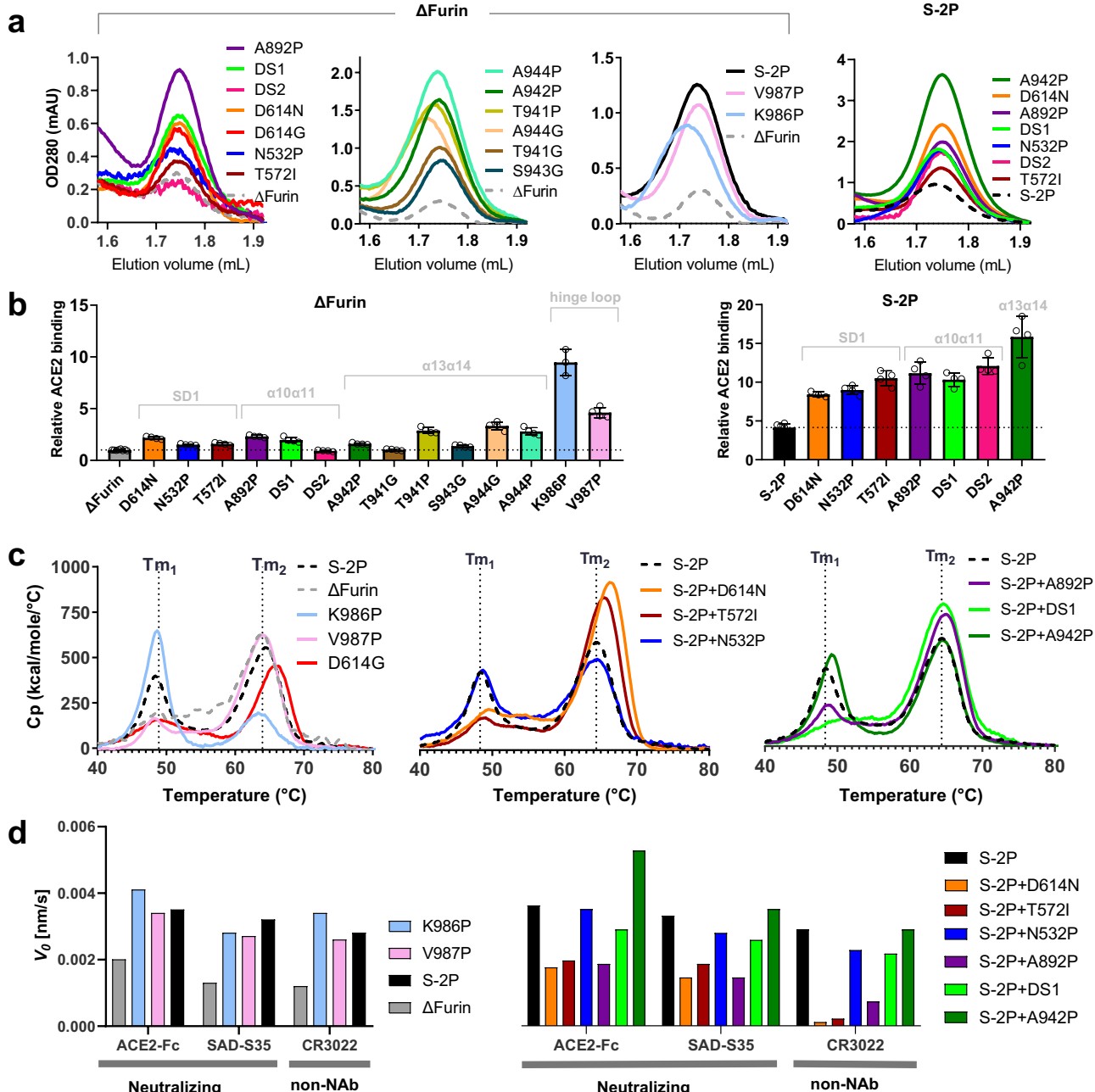

**Fig. 1 Characterization of SARS-CoV-2 S mutants containing single stabilizing mutations. a** Analytical SEC of S variants on an SRT-10C SEC-500 15 cm column showing the trimer peak (solid line) relative to the control, which is the un-mutated backbone (dashed line). **b** ACE2-Fc binding to S protein mutants based on AlphaLISA of ΔFurin variants (left panel) and S-2P (right panel). Data are represented as mean + SD of $n = 4$ biologically independent samples in one experiment. Mutants were grouped according to the structural regions indicated in light gray. **c** Temperature stability of purified S trimers as measured by DSC. Two melting events are indicated by Tm1 and Tm2. **c** (left panel) Uncleaved SARS2-S variants with furin site mutations (ΔFurin), with one stabilizing proline mutation in the hinge loop (ΔFurin K986P or ΔFurin V987P), and both proline mutations in the hinge loop (S-2P); (middle panel) ΔFurin variants with indicated mutations in S1 and (right panel) ΔFurin variants with indicated mutations in S2. **d** Binding of SAD-S35, ACE2, and CR3022 to purified S proteins measured with BioLayer Interferometry, showing the initial slope $V0$ at the start of binding. Source data are provided as a Source Data file.

(Supplementary Fig. 3), in agreement with a more compact structure. The largest shift was caused by D614N. Since 614 corresponds to a position in the S protein subject to the most extensive adaptation to the human host-D614G[32], this variant was also tested for expression and stability. The results show that the D614G change has a very similar effect as D614N. Both increase trimer yields (Fig. 1a), increase Tm by ~2 °C (Fig. 1c), and reduce SAD-S35, ACE2, and CR3022 binding substantially

(Supplementary Fig. 5). Interestingly, both D614G and D614N full-length variants show increased fusogenicity compared to wild-type D614 (Supplementary Fig. 4).

Next, we made variants with combinations of selected mutations to evaluate if their effects on expression, trimer closure, and stability are additive. Because of their positive effect on the stability of the spike and location in different protein domains, both D614N and A892P were selected. In addition, A942P was selected

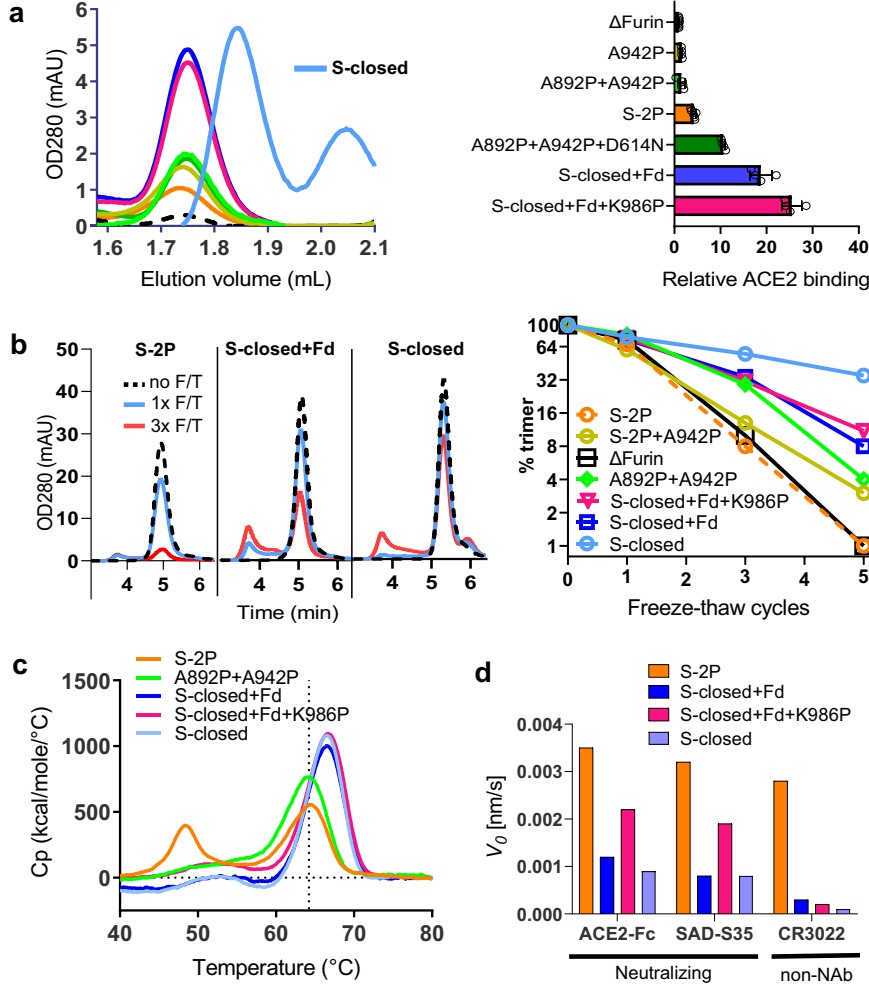

**Fig. 2 Characterization of SARS-CoV-2 S mutants containing combinations of stabilizing mutations. a** Analysis of cell culture supernatant after transfection using analytical SEC using an SRT-10C SEC-500 15 cm column (left panel) and ACE2-Fc binding based on AlphaLISA (right panel) of ΔFurin S with combinations of stabilizing mutations. Data are represented as mean + SD of $n = 4$ biologically independent samples in one experiment. **b** Freeze-thaw stability of purified uncleaved S trimers with indicated stabilizing mutations as measured by analytical SEC. Chromatograms are shown for non-frozen (dashed black line), 1×frozen (light blue line), and 3×frozen (red line). **c** Temperature stability of purified combined S trimer variants by DSC. **d** Binding of SAD-S35, ACE2, and CR3022 to purified combined S trimer variants measured with BioLayer Interferometry, showing the initial slope $V0$ at the start of binding. Source data are provided as a Source Data file.

for its strong enhancement of protein yields (Fig. 1a). These mutations were combined with hinge loop prolines resulting in two combos, a quadruple mutant S-closed + Fd containing D614N + A892P + A942P + V987P and a quintuple mutant S-closed + Fd + K986P. Although both combos showed a similar, approximately 5-fold improvement in yields compared to S-2P, the addition of K986P increased ACE2 binding with similar expression levels (Fig. 2a). Interestingly, the quadruple mutant in which the Fd trimerization domain is deleted (S-closed), showed a 6.4-fold improvement in yield compared to S-2P. Its trimer peak was shifted towards a longer retention time due to the smaller size which was confirmed by MALS analysis (Fig. 2a and Supplementary Table 2).

S-closed with and without Fd were further characterized for resilience during repeated freeze-thaw cycles (Fig. 2b and Supplementary Table 3). Analytical SEC showed that only 8% of the original S-2P trimer was present after 3, and only 1% after 5 cycles (Fig. 2b, right panel). The trimer content is significantly improved for S-closed, which retained 55% of intact trimers after 3, and 35% after 5 freeze-thaw cycles. All combos displayed higher thermal stability with only a single Tm$_{50}$ at about 66 °C

(Fig. 2c) and decreased levels of ACE2 and antibody binding compared to S-2P control (Fig. 2d and Supplementary Fig. 5).

The S-closed+Fd quadruple mutant was then imaged by cryo-EM. A 2-steps 3D classification illustrates that out of 833,000 classified particles, ~80% was closed with all RBDs in the down state and 38% was categorized into a well-defined closed class while ~20% showed 1 RBD-up (Supplementary Fig. 6). Further processing of the 320,000 closed conformation particles allowed us to obtain a 2.8 Å electron potential map for the closed conformation and a 3.0 Å electron potential map for the 1 RBD-up (one up) conformation (Supplementary Fig. 7). An atomic model that was built into the 2.8 Å electron potential map confirmed that S retains the prefusion spike conformation (Fig. 3). The NTD and RBD density is less defined than for the rest of the map, suggesting flexibility in these regions. The closed structure (see Fig. 3a) is highly reminiscent of the one previously solved by Walls et al.[26]. The two structures differ by 2.2 Å all-atom RMSD and there are no significant differences in terms of the relative position of domains or domain conformation. The RBD is relatively more defined and the NTD less defined compared to the closed trimer described by Walls et al. The stabilizing mutations

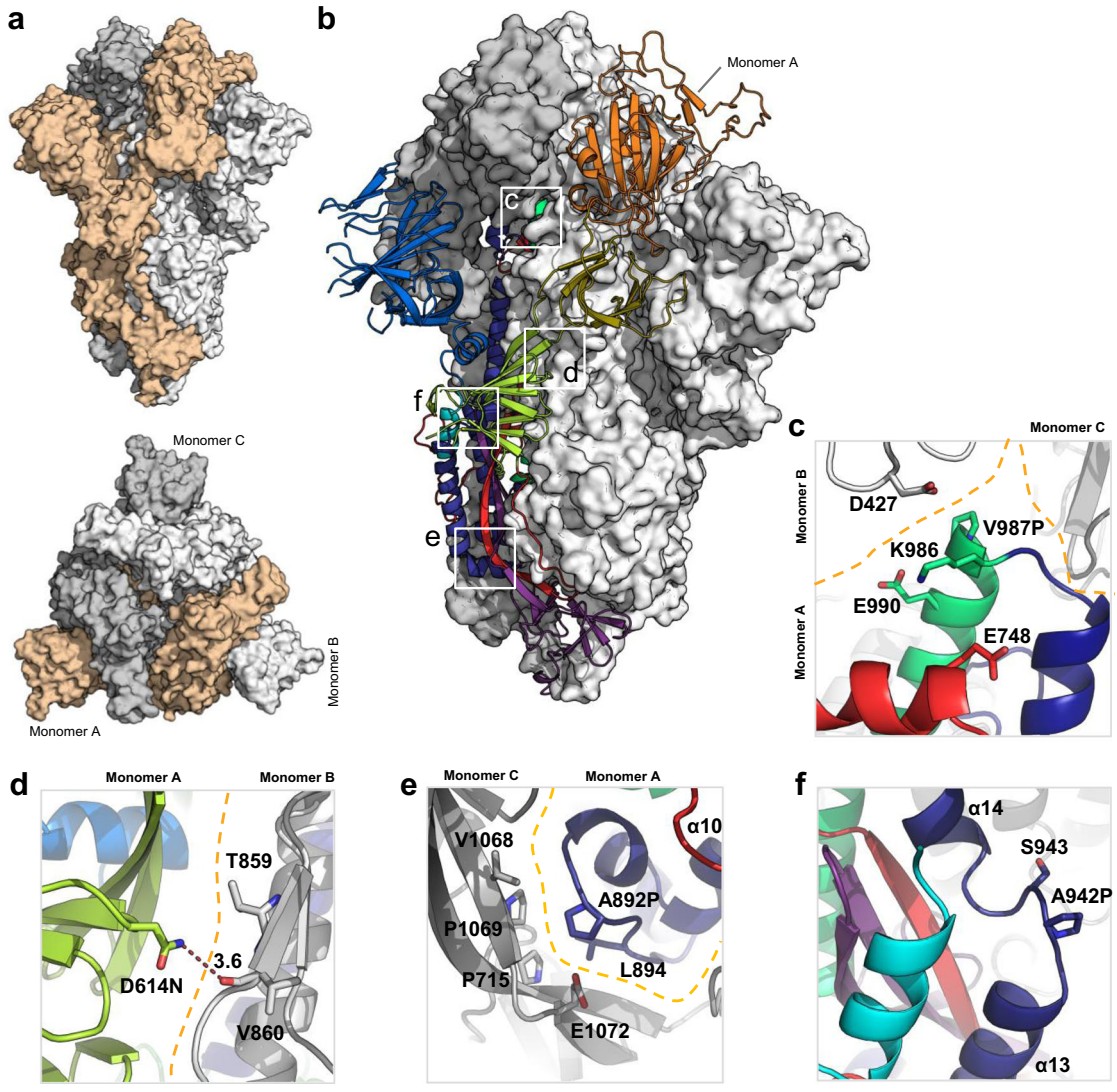

**Fig. 3 Structural characterization of S-closed-Fd. a** Cryo-EM structure of the most abundant trimer class of S-closed-Fd-a closed S protein trimer (PDB accession code: 7A4N). Monomers are colored in light orange, white and gray and the spike is shown from the side (upper panel) and from the top (lower panel) view. **b** S-closed-Fd trimer with monomer A plotted as cartoon. Structural domains are colored according to the same color code as in Supplementary Fig. 1. Areas where stabilizing mutations are introduced are indicated with white squares. **c–f** Each of the four single-point mutations introduced in S-closed-Fd is shown in detail. Domains of the new structure are colored according to the same color code as used in Supplementary Fig. 1. The boundary between monomers has been additionally indicated with an orange dashed line where applicable. Secondary structure α10, α13, and α14 are indicated in panels (**e** and **f**).

do not significantly affect the backbone conformation of the closed trimer.

## Discussion

SARS-CoV-2 S protein is unstable[9,12,23–27] and although the introduction of a double proline (K986P and V987P) in the hinge loop at the C-terminus of HR1 was shown to improve the stability of the prefusion conformation[9,26], the S protein still suffered from instability (Figs. 1c and 2b). We show that while each proline mutation increases trimer expression, the K986P variant also reduces the interaction with the S1 head, releasing the RBD to the 'up' configuration based on increased ACE2 and CR3022 binding (Figs. 1d and 2d) and a leftward shift in the SEC profile (Fig. 1a). The effect is partially compensated by the introduction of the second stabilizing mutation—V987P. We describe here two groups of substitutions that further stabilized the S-2P prefusion trimers and reduced RBD exposure. The first group of stabilizing mutations was identified in the S2 HR1 region that undergoes an

extensive conformational change during fusion. Introduction of stabilizing proline and glycine mutations in loops α10α11 and α13α14 and disulfides in loop α10α11 and between the central helix and C-terminus of HR1 resulted in significantly increased trimer yields. It is likely that the mutations in S2 facilitate folding of the protein during expression by fixing loop conformations with otherwise intrinsic alpha-helical propensity that is necessary for driving the fusion conformational change. This strategy greatly improved prefusion S protein expression levels (Fig. 1a). Recently, Hsieh et al., in parallel to our study, demonstrated the stabilizing effect of the A942P and A892P mutations[12]. Mutations identified in the head region clustered near the SD1 subdomain. Interestingly, introduction of two of these mutations (D614N and T572I) into the S-2P backbone resulted, apart from increased expression, in a striking improvement of thermal stability (Fig. 1c) and very low RBD exposure (Fig. 1d).

Based on the biochemical characterization of single-point mutants and their subdomain placement we selected four

substitutions with low surface exposure (D614N, A892P, A942P, and V987P) to create S-closed. This construct does not contain a heterologous trimerization domain and exhibits a 6.4-fold increase in expression (Fig. 2a), high thermal and freeze–thaw stability (Fig. 2b, c), and antigenicity reminiscent of a closed trimer (Fig. 2d). Both D614G and D614N variants show increased fusogenicity and stability (Fig. 1c and Supplementary Fig. 4), which may be explained by a decrease in premature shedding of S1[33]. The interaction of D614 with the fusion peptide proximal region (FPPR) (residues 828–853)[27] may play a role in stabilizing the spike during expression, as recently elucidated by a low pH structure[34], in which D614 is in direct proximity of D839 and forms a hydrogen bond with the carbonyl of V860, suggesting aspartate protonation. In our new structure, N614 forms two interprotomeric hydrogen bonds with the sidechain of T859 and backbone carbonyl of V860 (Fig. 3b), effectively mimicking the stabilizing effect of the protonated aspartate residue, without the pH dependence. It remains unknown why the D614G has a similar positive effect on stability, but perhaps the high cluster of negative charges in the interface between S1 and S2 destabilizes the protein, and the 614N and 614G mutations reduce this repulsion. None of the stabilizing proline mutations modify the local structure of the protein, while A892P adds new inter-protomeric interactions with the neighboring hybrid sheet, composed of both S1 and S2 strands (Fig. 3b). The stabilizing mutations and presence of K986, which interacts with Asp427 in the RBD of the neighboring monomer and Glu748 in S2 (Fig. 3b), allow the S trimer to maintain predominantly the closed prefusion configuration.

The viral spike is mostly closed[28] and structurally very similar to other known closed S-2P spike conformations[25,26] and especially the closed wild-type structure of Xiong et al.[25], with an all-atom RMSD of 1.7 Å. The FPPR is not visible as in the so-called locked S structure of Xiong et al.[25] or the structure of Cai et al.[27], both showing tighter packing of the head domains against S2. Although many neutralizing antibodies are directed against the RBD, the antibodies with the highest neutralizing activity bind the RBD in the down state[4,6,13]. Similar to other class I fusion proteins, a closed conformation may be more reminiscent of a transmitted virus, and antibodies that recognize the closed state may be more important for protection as has been shown for HIV[35]. Furthermore, the closed conformation can potentially induce antibodies with higher cross-reactivity since the downstate surface is more conserved. A stable closed S trimer with minimal non-exposed mutations and without a Fd that shows a significant increase in expression levels may advance the development of novel (subunit) vaccine immunogens and further improve genetic vaccines, diagnostics, or isolation of antibodies.

## Methods

**Protein expression and purification.** A plasmid corresponding to the semi-stabilized SARS-CoV-2 S-2P protein[9] was synthesized, codon-optimized, cloned into pCDNA2004, and sequenced at GenScript (Piscataway, NJ 08854), where also all the variants with different amino acid substitutions were generated. A variant with a HIS-tag and a variant with a C-tag were purified. The expression platform used was the Expi293F cells. The cells were transiently transfected using Expi-Fectamine (Life Technologies) according to the manufacturer's instructions and cultured for 6 days at 37 °C and 10% CO$_2$. The culture supernatants were harvested and spun for 5 minutes at 300 × g to remove cells and cellular debris. The spun supernatants were subsequently sterile filtered using a 0.22 µm vacuum filter and stored at 4 °C until they were purified within 1–2 days of harvest. HIS-tagged SARS-CoV-2 S trimers were purified using a two-step purification protocol by 1- or 5-ml complete HIS-tag columns (Roche). Proteins were further purified by size-exclusion chromatography using a HiLoad Superdex 200 16/600 column (GE Healthcare).

**Antibodies and reagents.** SAD-S35 was purchased at Acro Biosystems. ACE2-Fc was made according to Liu et al. (2018)[36], *Kidney international*. For CR3022, the heavy and light chains were cloned into a single IgG1 expression vector to express a fully human IgG1 antibody. CR3022 was made by transfecting the IgG1 expression construct using the ExpiFectamine™ 293 Transfection Kit (ThermoFisher) in Expi293F (ThermoFisher) cells according to the manufacturer's specifications. CR3022 was purified from serum-free culture supernatants using mAb Select SuRe resin (GE Healthcare) followed by rapid desalting using a HiPrep 26/10 Desalting column (GE Healthcare). The final formulation buffer was 20 mM NaAc, 75 mM NaCl, and 5% Sucrose pH 5.5.

**Differential scanning calorimetry (DSC).** Melting temperatures for S protein variants were determined using a PEAQ-DSC system. In all, 325 µl of 0.3 mg/ml protein sample was used per measurement. The measurement was performed with a start temperature of 20 °C and a final temperature of 100 °C at a scan rate 100 °C/h and the feedback mode; Low (= signal amplification).

**Differential scanning fluorometry (DSF).** A total of 0.2 mg of purified protein in 50 µl PBS pH 7.4 (Gibco) was mixed with 15 µl of 20 times diluted SYPRO orange fluorescent dye (5000×stock, Invitrogen S6650) in a 96-well optical qPCR plate. A negative control sample containing the dye only was used for reference subtraction. The measurement was performed in a qPCR instrument (Applied Biosystems ViiA 7) using a temperature ramp from 25 to 95 °C with a rate of 0.015 °C per second. Data were collected continuously. The negative first derivative was plotted as a function of temperature. The melting temperature corresponds to the lowest point in the curve.

**BioLayer Interferometry (BLI).** A solution of SAD-S35 at a concentration of 0.5 µg/ml and ACE2-Fc and CR3022 at a concentration of 10 µg/ml was used to immobilize the ligand on anti-hIgG (AHC) sensors (FortéBio, cat. #18–5060) in 1×kinetics buffer (FortéBio, cat. #18–1092) in 96-well black flat-bottom poly-propylene microplates (FortéBio, cat. #3694). The experiment was performed on an Octet RED384 instrument (Pall-FortéBio) at 30 °C with a shaking speed of 1000 rpm. Activation was 600 s, immobilization of antibodies 900 s, followed by washing for 600 s, and then binding the S proteins for 300 s. The data analysis was performed using the FortéBio Data Analysis 12.0 software (FortéBio).

### Cryo-EM

*Cryo-EM Grid Preparation and Data Collection.* SARS-CoV-2 S protein samples were prepared in 20 mM Tris, 150 mM NaCl, pH 7 buffer at a concentration of 0.15 mg/ml and applied to glow discharged Quantifoil R2/2 200 mesh grids before being double side blotted for 3 s in a Vitrobot Mark IV (Thermo Fisher Scientific) and plunge frozen into liquid ethane cooled. Grids were loaded into a Titan Krios electron microscope (Thermo Fisher Scientific) operated at 300 kv, equipped with a Gatan K3 BioQuantum direct electron detector. A total of 9760 movies were collected over two microscopy sessions at The Netherlands Center for Electron Nanoscopy (NeCEN). Detailed data acquisition parameters are summarized in Supplementary Table 4.

*Cryo-EM image processing.* Collected movies were imported into RELION-3.1-beta[37] and subjected to beam-induced drift correction using MotionCor2[38] and CTF estimation by CTFFIND-4.1.18[39]. Detailed steps of the image processing workflow are illustrated in Supplementary Figure 7. Final reconstructions were sharpened and locally filtered in RELION post-processing.

*Model building and refinement.* The SARS-CoV-2 S PDBID 6VXX and 6VSB structures[9,26] were used as starting models. PHENIX-1.18.261[40], Coot[41], and the Namdinator webserver[42] were iteratively used to build atomic models. Geometry and statistics are given in Supplementary Table 4 and Supplementary Table 5. Final maps were displayed using UCSF ChimeraX[43].

**AlphaLISA.** Crude supernatants were diluted 300 times in AlphaLISA buffer (PBS + 0.05% Tween-20 + 0.5 mg/mL BSA). Then, 10 µl of each dilution was transferred to a 96-well plate and mixed with 40 µl acceptor beads, donor beads, and ACE2-Fc. The donor beads were conjugated to ProtA (Cat. #AS102M, Perkin Elmer), which binds to ACE2Fc. The acceptor beads were conjugated to an anti-Flag antibody (Cat. #AL112M, Perkin Elmer), which binds to the Flag-tag of the construct. The mixture of the supernatant containing the expressed S protein, the ACE-2-Fc, donor beads, and acceptor beads was incubated at room temperature for 2 h without shaking. Subsequently, the chemiluminescent signal was measured with an Ensight plate reader instrument (Perkin Elmer). The average background signal attributed to mock-transfected cells was subtracted from the AlphaLISA counts. Subsequently, the whole data set was divided by the signal measured for the SARS CoV-2 S protein having the S backbone sequence signal to normalize the signal for each of the S variants tested to the backbone.

**Analytical SEC.** An ultra-high-performance liquid chromatography system (Vanquish, Thermo Scientific) and µDAWN TREOS instrument (Wyatt) coupled to an Optilab µT-rEX Refractive Index Detector (Wyatt), in combination with an in-line Nanostar DLS reader (Wyatt), was used for performing the analytical SEC experiment. The cleared crude cell culture supernatants were applied to an

SRT-10C SEC-500 15 cm column (Sepax Cat. #235500–244615) with the corresponding guard column (Sepax) equilibrated in running buffer (150 mM sodium phosphate, 50 mM NaCl, pH 7.0) at 0.35 mL/min. When analyzing supernatant samples, μMALS detectors were offline and analytical SEC data were analyzed using Chromeleon 7.2.8.0 software package. The signal of supernatants of non-transfected cells was subtracted from the signal of supernatants of S transfected cells. When purified proteins were analyzed using SEC-MALS, μMALS detectors were inline and data were analyzed using Astra 7.3 software package. For the protein component, a dn/dc (mL/g) value of 0.1850 was used and for the glycan component a value of 0.1410. Molecular weights were calculated using the RI detector as [C] source and mass recoveries using UV as [C] source.

**Cell–cell fusion assay.** Quantitative cell–cell fusion assays were performed to ascertain the relative fusogenicity of the different D614 S protein variants by using the NanoBiT complementation system (Promega). Donor HEK293 cells were transfected with full-length S and the 11 S subunit in 96-well white flat-bottom TC-treated microtest assay plates. Acceptor HEK293 cells were transfected in six-well plates (Corning) with ACE2, TMPRSS2, and the PEP86 subunit, or just the PEP86 subunit ('Mock') as a negative control. All proteins were expressed from pcDNA2004 plasmids using Trans-IT transfection reagent according to the manufacturer's instructions. Eighteen hours after transfection, the acceptor cells were released by 0.1% trypsin/EDTA and added to the donor cells at a 1:1 ratio for 4 h. Luciferase complementation was measured by incubating with Nano-Glo® Live Cell Reagent for 3 m, followed by read-out on an Ensight plate reader (PerkinElmer).

## Data availability
The authors declare that the main data supporting the findings of this study are available within the article and its Supplementary Information files. Some data that support the findings of this study are deposited in the Protein Data Bank database with accession codes 7A4N and 7AD1. Source data are provided with this paper.

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

## Acknowledgements
This work benefited from access to the Netherlands Centre for Electron Nanoscopy (NeCEN) at Leiden University, an Instruct-ERIC center with assistance from Wen Yang and Frederic Bonnet. We thank Wouter Koudstaal for advice and assistance. We thank Ilona Bisschop, Martijn de Man, Ava Sadi, Anne-Marie de Gooyert, Lam Le, and Annemart Koornneef for technical support. We thank Xiaodi Yu and Sujata Sharma for support with structure refinement.

## Author contributions

J.J., L.R., M.J.G.B. and J.P.M.L. designed the study. J.J., L.R. and J.P.M.L. performed structure-based design of mutations. T.R., S.B., P.B. and R.V. planned and/or performed biochemical assays and purifications. L.L.R.R. performed EM sample preparation, data collection, data processing, and analysis. J.J., L.R., M.J.G.B., L.L.R.R. and J.P.M.L. wrote the paper.

## Competing interests

J.J., L.R., M.J.G.B. and J.P.M.L. are co-inventors on related vaccine patent applications. J.J., L.R., S.B., P.B., R.V., T.R., M.J.G.B. and J.P.M.L. are employees of Janssen Vaccines & Prevention B.V.; L.R., J.J., and J.P.M.L. hold stock of Johnson & Johnson.
