## [Peer Review File · Nature Communications]

Reviewer Comments first round:

Reviewer #1 (Remarks to the Author):

This manuscript describes the identification of a number of point mutations in the SARS-CoV-2 spike glycoprotein that increase its thermostability and decrease the dynamic behavior of the receptor binding domain in its "down", or closed, conformation. The study was very carefully done and the data provided are of very high quality, including the 2.8 Å resolution of the stabilized closed form obtained by cryo-EM. Although several results are available already in the literature about mutations that stabilize the S protein, this paper goes further, as it finds a way of eliminating the stabilization foldon. Importantly, one of the stabilizing mutations is at position 614 of S1, which has been shown to occur naturally and has been fixed by now in the virus circulating in the human population, suggesting that it is a possible adaptation of the virus to facilitate human-to-human transmission.

Overall, this is a very important contribution to the design of stable immunogens to use in third generation vaccines against covid-19. I only have minor comments, which the authors may want to address.

1- The effect of the D614G spike variant appears to have been added at the last moment, and is only in the supplementary materials. It would be important to bring it into the main Figures in the revised version, along with the D614N mutant.

2- It would be interesting in the Discussion to refer to the structure published by Cai et al. (DOI: 10.1126/science.abd4251) on the full length S protein in the absence of any mutation. The authors found a close conformation there in which the NTD was ordered, as well as the fusion peptide proximal region (FPPR). What aspects seen in that structure are similar and what not, compared to the structure of the closed form with foldon described here. Similarly, it would be good to describe the relations to the "closed" conformation observed by Xiong et al. (<https://doi.org/10.1101/2020.06.15.152835>).

Minor comments:

1- In the legend to Figure 1 (line 95): What is main by "backbone"? Does it mean that the dashed line corresponds to a control, such as the un-mutated protein?

2- In line 112, I supposed that "Amongst the S1 mutants" is meant, not S2

3- Figure 2: I'm puzzled by the discrepancy seen in the poor reactivity of the S-2P variant in 2A, right panel, when compared to the data shown in 2D, where the S-2P appears as most reactive to ACE2 and the antibodies. Can this be clarified?

4- Line FPPR: please specify the aa range of the FPPR, this will help the reader to know if the positions discussed later in the paragraph (V860, for instance) fall in the FPPR or not.

Reviewer #2 (Remarks to the Author):

In the manuscript entitled "Stabilizing the Closed SARS-CoV-2 Spike Trimer" the authors perform structure-guided stabilization of the SARS-CoV-2 spike. Working in the context of the S ectodomain the authors introduced mutations in the HR1 region of the S2 subunit and the SD1 at the

interface with S2 to create a variant (S-closed) that expressed at higher levels than the commonly used S-2P construct, and importantly, no longer required a heterologous trimerization domain to retain the integrity of its 3-dimensional structure. The authors characterize the differently stabilized constructs using a variety of biophysical and biochemical measures, including size exclusion chromatography, DSC, DSF, receptor and antibody binding assays. Using cryo-EM the authors determine the structure of the "S-closed+Fd" molecule (ie., S-closed construct with the foldon trimerization domain intact), and show 80:20 population distribution between the all-RBD down and 1-RBD up spikes. The authors demonstrate reduced binding of the "S-closed" molecule to ligands that require an up RBD configuration.

Major comments:

- (1) This study adds to the growing body of literature on stabilization of the SARS-Cov-2 S-ectodomain. The most significant advance in this study is the creation of a stabilized S ectodomain that no longer depends upon a heterologous trimerization motif for structural integrity. This new molecule with improved expression yields has potential use in vaccine development.
- (2) The cryo-EM studies appear well-done with clear presentation of the workflow.
- (3) One major area of concern and potential weakness in this study is in the integrity of the various samples used for biophysical and antigenic measurements. The authors state the following in the methods while describing the purification of the S constructs – “The culture supernatant was harvested and spun for 5 minutes at 300 g to remove cells and cellular debris. The spun supernatant was subsequently sterile-filtered using a 0.22 µm vacuum filter and stored at 4°C until use.” In light of multiple recent reports from different labs (<https://www.nature.com/articles/s41594-020-0478-5>, <https://www.biorxiv.org/content/10.1101/2020.07.12.199588v2>) demonstrating denaturation of S ectodomain constructs upon storage at 4C, it is critical that the authors address this aspect of their work because it will very likely impact the results presented, especially for the less stabilized constructs. It is possible, in fact even likely, that the stabilized constructs are no longer sensitive to denaturation upon storage at 4C, thereby the final conclusions and results of the paper may be fine. Yet, because a significant proportion of the manuscript is devoted to describing the properties of less-stabilized constructs and comparing them with each other, it is critical that these studies be performed with greater rigor. Otherwise, it is difficult to judge whether the differences in properties demonstrated in Figures 1 and 2, for example, are arising due to differences between the constructs, or reflect how long they have been stored at 4C, or how each construct is responding to storage at 4C. To ensure reproducibility of the data presented in this paper the authors should provide details on how long each protein was stored at 4C, and if the storage times were not equivalent for each construct, the measurements should be repeated. It is highly recommended that the key measurements presented be repeated with samples that have not been stored at 4C for any length of time.
- (4) Statistics:
 - a. Figure 1B and 2A: Error bars are shown. Authors should report number and nature of replicates (technical replicates, independent protein samples...)
 - b. Figure 1D and 2D: Can the authors add error bars here?
 - c. Figure S2: Can the authors include replicate measures here?
 - d. Figure S5: legend should describe how the errors were calculated?

Minor comments:

- (1) Some of the figure legends appear incomplete, for example, S5 and S6.
- (2) The cryo-EM workflow (Figure S7) needs more labels describing each step in the workflow.
- (3) Table S3 needs better description of how the numbers shown in the table are derived.
- (4) The model refinement statistics look fine, although the clashscore can possibly be improved by employing newer implementations such as in Isolde and Rosetta.
- (5) Figure 3 is captioned: Structural characterization of S-closed. But the structure presented is for the “S-closed+Fd” construct.

Priyamvada

Priyamvada Acharya, Ph.D.
Associate Professor
Director, Division of Structural Biology
Duke Human Vaccine Institute
Department of Surgery
Duke University School of Medicine
LSRC Wing A, Rm 06, Box 103020,
Durham, NC 27710
Ph: 919-660-5446
priyamvada.acharya@duke.edu

Reviewer #3 (Remarks to the Author):

Langedijk and colleagues describe the design and biophysical and structural characterization of a number of stabilized prefusion SARS-CoV-2 S trimer variants. They ultimately settled on a variant combining several mutations that is stabilized in the prefusion, 3-down (although not completely) conformation and expresses at higher levels than previous constructs even without a heterologous trimerization domain. Given the extreme impact of SARS-CoV-2 on the world and the potential utility of these new S variants in vaccine development, these results are highly significant and will be of interest to a broad audience. The work is well-laid out and the experiments straightforward and well-designed. The quality of the data is high. I recommend that the article be published in Nat Comms after minor revision.

Major comments:

My only major comment is that the text could use some polishing; there are many inconsistencies in formatting, editing, and syntax.

Minor comments:

-- Page 5: The authors should clarify that they measure trimer content by analytical SEC. They should further clarify, using a few words in the main text what the nature of the samples were that were loaded onto the column. This information is provided in the materials & methods, but it would help the reader to provide it very briefly in the main text as well.

-- Fig 1A: For all SEC figures, please use Elution volume (mL) instead of Time (min) on the x axis. Also, please specify in the figure caption the column that was used in these experiments.

-- Fig S2: The y axis should be changed to indicate that the derivatives of the fluorescent signal are plotted. Please also include the raw fluorescence signal (not the derivative) for completeness.

-- Page 7 / Fig 1D: Please use a few words to clarify what assay was used to measure antibody binding. Also, while the initial rates of increase of BLI signal may be an appropriate measure of relative binding, the raw BLI curves must be provided as supplementary material to allow the reader to interpret the data on their own. Also, the y axis in Fig 1D should be relabeled something like V_0 (nm/s).

-- Fig 2A: The authors should provide ACE2 binding data for S-closed. Also, there is no need for scientific notation on the x axis tick labels.

-- Fig S6: The authors should indicate in the caption that the data pertain to Fig 2D.

-- Page 10 / Fig 3: The authors should provide a slightly more detailed description of their structure of S-closed. What is the RMSD of their model compared to that of Walls et al.? Are there any regions that deviate significantly? Are there backbone-level changes near the mutations, or only changes in side chain configurations? Also, the Fig 3 caption should state whether the semi-transparent surface representations are actual density or are calculated from the model. These representations should be consistent across all four sub-panels (currently missing from the D614 panel). Also, mutations should be indicated as such using the residue labels (e.g., A942P), while non-mutated residues should be labeled as they currently are (e.g., V860).

-- Page 12, line 189: Other groups have described this instability as well, and their work should be cited here.

REVIEWER COMMENTS

Reviewer #1

“This manuscript describes the identification of a number of point mutations in the SARS-CoV-2 spike glycoprotein that increase its thermostability and decrease the dynamic behavior of the receptor binding domain in its “down”, or closed, conformation. The study was very carefully done and the data provided are of very high quality, including the 2.8 Å resolution of the stabilized closed form obtained by cryo-EM. Although several results are available already in the literature about mutations that stabilize the S protein, this paper goes further, as it finds a way of eliminating the stabilization foldon. Importantly, one of the stabilizing mutations is at position 614 of S1, which has been shown to occur naturally and has been fixed by now in the virus circulating in the human population, suggesting that it is a possible adaptation of the virus to facilitate human-to-human transmission.

Overall, this is a very important contribution to the design of stable immunogens to use in third generation vaccines against covid-19. I only have minor comments, which the authors may want to address.”

1. *“The effect of the D614G spike variant appears to have been added at the last moment, and is only in the supplementary materials. It would be important to bring it into the main Figures in the revised version, along with the D614N mutant.”*

We added the analytical SEC data of the D614G mutant in Figure 1A (see red curve) and we added the DSC data of the D614G mutant in Figure 1C (also red curve). Unfortunately, we do not have the DSC data of the D614N in the Δ Furin background and we do not have it for the D614G in the S-2P background. As a result Supplementary Figure 4 was removed, and all following figures were renumbered.

We also modified the main text on page 7, lines 116-118: “Amongst the S12 mutants (Fig. 1C central panel, Table S1) D614N and the natural variant D614G diminished the peak height of the first Tm50 and increased the second by almost 2°C in their respective backgrounds, which was subsequently confirmed with differential scanning fluorimetry DSF (Fig. S2).”

2. *“It would be interesting in the Discussion to refer to the structure published by Cai et al. (DOI: 10.1126/science.abd4251) on the full length S protein in the absence of any mutation. The authors found a close conformation there in which the NTD was ordered, as well as the fusion peptide proximal region (FPPR). What aspects seen in that structure are similar and what not, compared to the structure of the closed form with foldon described here. Similarly, it would be good to*

describe the relations to the “closed” conformation observed by Xiong et al. (<https://doi.org/10.1101/2020.06.15.152835>).”

We added in the discussion section page 14/15 lines 242-245: “The viral spike is mostly closed²⁸ and structurally very similar to other known closed S-2P spike conformations^{25,26} and especially the closed wild-type structure of Xiong et al²⁵, with all atom RMSD of 1.7Å. The FPPR is not visible as in the so called locked S structure of Xiong et al²⁵ or the structure of Cai et al²⁷, both showing tighter packing of the head domains against S2.”

Minor comments:

1. “In the legend to Figure 1 (line 95): What is main by “backbone”? Does it mean that the dashed line corresponds to a control, such as the un-mutated protein?”

Yes, that is correct. We added the underline part in line 99 (previously line 95): “relative to the control, which is the un-mutated backbone (dashed line)”.

2. “In line 112, I supposed that “Amongst the S1 mutants” is meant, not S2”

Yes, this is correct. Thanks for this catch, we corrected the mistake (now in line 116).

3. “Figure 2: I’m puzzled by the discrepancy seen in the poor reactivity of the S-2P variant in 2A, right panel, when compared to the data shown in 2D, where the S-2P appears as most reactive to ACE2 and the antibodies. Can this be clarified?”

The reason for this apparent discrepancy lies in the fact that the data shown in figure 2A was obtained with unpurified cell culture supernatants and in figure 2D was done with purified proteins. In the supernatants the S protein (trimer) concentration varies for each variant and is much lower for the S-2P variant than the more stabilized variants, which explain the lower ACE2 binding in Figure 2A. For figure 2D the S protein concentrations are the same.

To clarify that the ACE2 binding in the supernatant is dependent on the trimer yield seen in the left panel of Figure 2A, we added the underlined part to sentence on page 8, line 149 “Although both combos showed similar, approximately 5-fold improvement in yields compared to S-2P, the addition of K986P increased ACE2 binding with similar expression levels (Fig. 2A).”

4 “Line FPPR: please specify the aa range of the FPPR, this will help the reader to know if the positions discussed later in the paragraph (V860, for instance) fall in the FPPR or not.”

We added on page 14 line 229: “(residues 828-853)” and the reference to the paper (Cai et al) in which the FPPR region was first mentioned.

Reviewer #2

“In the manuscript entitled "Stabilizing the Closed SARS-CoV-2 Spike Trimer" the authors perform structure-guided stabilization of the SARS-CoV-2 spike. Working in the context of the S ectodomain the authors introduced mutations in the HR1 region of the S2 subunit and the SD1 at the interface with S2 to create a variant (S-closed) that expressed at higher levels than the commonly used S-2P construct, and importantly, no longer required a heterologous trimerization domain to retain the integrity of its 3-dimensional structure. The authors characterize the differently stabilized constructs using a variety of biophysical and biochemical measures, including size exclusion chromatography, DSC, DSF, receptor and antibody binding assays. Using cryo-EM the authors determine the structure of the “S-closed+Fd” molecule (ie., S-closed construct with the foldon trimerization domain intact), and show 80:20 population distribution between the all-RBD down and 1-RBD up spikes. The authors demonstrate reduced binding of the “S-closed” molecule to ligands that require an up RBD configuration.”

Major comments:

(1) *“This study adds to the growing body of literature on stabilization of the SARS-Cov-2 S ectodomain. The most significant advance in this study is the creation of a stabilized S ectodomain that no longer depends upon a heterologous trimerization motif for structural integrity. This new molecule with improved expression yields has potential use in vaccine development.”*

(2) *“The cryo-EM studies appear well-done with clear presentation of the workflow.”*

(3) *“One major area of concern and potential weakness in this study is in the integrity of the various samples used for biophysical and antigenic measurements. The authors state the following in the methods while describing the purification of the S constructs – “The culture supernatant was harvested and spun for 5 minutes at 300 g to remove cells and cellular debris. The spun supernatant was subsequently sterile-filtered using a 0.22 μ m vacuum filter and stored at 4°C until use.” In light of multiple recent reports from different labs (<https://www.nature.com/articles/s41594-020-0478-5>, <https://www.biorxiv.org/content/10.1101/2020.07.12.199588v2>) demonstrating denaturation of S ectodomain constructs upon storage at 4C, it is critical that the authors address this aspect of their work because it will very likely impact the results presented, especially for the less stabilized constructs. It is possible, in fact even likely, that the stabilized constructs are no longer sensitive to denaturation upon storage at 4C, thereby the final conclusions and results of the paper may be fine. Yet, because a significant proportion of the manuscript is devoted to describing the properties of less-stabilized constructs and comparing them with each other, it is critical that these studies be performed with greater rigor. Otherwise, it is difficult to judge whether the differences in properties demonstrated in Figures 1 and 2, for example, are arising due to differences between*

the constructs, or reflect how long they have been stored at 4C, or how each construct is responding to storage at 4C. To ensure reproducibility of the data presented in this paper the authors should provide details on how long each protein was stored at 4C, and if the storage times were not equivalent for each construct, the measurements should be repeated. It is highly recommended that the key measurements presented be repeated with samples that have not been stored at 4C for any length of time.”

The reviewer makes a very good point. We made sure we treated the samples as equally as possible, but as the reviewer mentioned, the 4°C sensitivity does not apply to the stabilized variants. We do not have concern about the intermediate variants either, as the supernatants were all tested in parallel and had all the same incubation time and treatment. Since they could not be purified in parallel, we did all we could to treat them the same way. In a parallel effort we were able to purify the 16 proteins in two days. Ten proteins were purified on the first day and the remaining six on the second day. The group of 10 proteins was harvested at day 0. The other 6 remained in the supernatant in the fridge, and 10 were directly purified and stored in the fridge for one day. The six were purified after one day of storage at 4°C, purified and kept in the fridge also for one day. In summary, all proteins were tested in Octet and DSF after 2 days at 4°C. Some of the proteins were measured in Octet on two consecutive days, showing very little variation (see answer to question b about statistics), so storage of purified proteins in the fridge for one day or two days does not seem to make a difference. Maybe the instability at 4°C occurs within the first one or two days. Therefore, we are confident that with the data we show the different constructs can be compared very well with each other.

(4) Statistics:

a. *“Figure 1B and 2A: Error bars are shown. Authors should report number and nature of replicates (technical replicates, independent protein samples...)”*

We corrected “± SEM” with “+ SD (n=2) of two independent samples”

b. *“Figure 1D and 2D: Can the authors add error bars here?”*

Unfortunately, we do not have error bars for each sample because we only measured on two consecutive days for a subset of constructs (see figure under this paragraph). For samples that were tested twice, the error bars are not large (+ SD, n=2 technical replicates). We decided not to show error bars for a subset of samples, since that could be misleading the readers into thinking the error is very low for the ones without a replicate. We believe the error bars do not change the conclusions since the trends are very clear with the panel of antibodies we used.

c. *“Figure S2: Can the authors include replicate measures here?”*

We now added two more curves to the figures and adjusted the legend. In the previous legend it said that the measurement was done in duplicate, but in fact it was done in triplicate. The calculations of the melting temperatures were based on the averages of the three measurements.

d. *“Figure S5: legend should describe how the errors were calculated?”*

We have expanded the legend of Figure S4 (former Figure S5). The legend now included information on how the errors were calculated: “Plotted is the mean +SD, data represents the average of two separate measurements”

Minor comments:

(1) *“Some of the figure legends appear incomplete, for example, S5 and S6.”*

We completed the figure legend of Figure S4 (former Figure S5). The full legend now says: “Quantitative cell-cell fusion assay in HEK293 cells using the NanoBiT split-luciferase complementation system. Donor cells were transfected with full-length wildtype S, or variants thereof, and the 11S luciferase subunit, acceptor cells were transfected with ACE2, TMPRSS2 and the PEP86 luciferase subunit, or just the PEP86 luciferase subunit (‘Mock’) as negative control. After 18 hr donor and acceptor cells were mixed at a 1:1 ratio and incubated for 4 hr at 37C, at

which time the luciferase signal was measured. Plotted is the mean +SD, data represents the average of two separate measurements. The level of fusion measured for wildtype S is indicated with a dotted line.”

We also expanded Figure S5 (former Figure S6), including the legend, which now says: “Bio-Layer Interferometry curves of SARS-CoV-2 S protein mutants and combinations binding to ACE2-Fc fusion, SAD-S35 and CR3022 antibodies. The curves were used to calculate the initial slope V_0 plotted in Figures 1D and 2D. In the upper three panels, S-2P and Δ Furin backgrounds are plotted in black and grey respectively. In the remaining panels, only S-2P curves are plotted for comparison.”

(2) *“The cryo-EM workflow (Figure S7) needs more labels describing each step in the workflow.”*

The cryo EM workflow Figure was adjusted and the labels were added.

(3) *“Table S3 needs better description of how the numbers shown in the table are derived.”*

We added “Areas under the curves of the trimer peaks in SEC for non-frozen samples (0x FT) and after multiple cycles of freeze-thaw (1x, 3x, and 5x FT), normalized to their respective non-frozen samples (100%).” to clarify. We also removed the corresponding line from the main text, page 11, line 166-168: “The areas under the curves (AUCs) of the trimer peaks in SEC were set to 100% for the non frozen sample, which dropped after freeze-thawing (Table S3).”

(4) *“The model refinement statistics look fine, although the clashscore can possibly be improved by employing newer implementations such as in Isolde and Rosetta.”*

We have performed additional refinement. To further improve the model geometry and density fitting, hydrogens were added to the model in the last round of real space refinement using PHENIX and followed by 2 rounds of manual rebuilding. The final models were analyzed with MolProbity, EMRinger. We managed to decrease the clash-score down to 5.08 for the closed structure and 8.72 for the open structure. In the process other validation and Ramachandran parameters had to be updated, but they remained in the acceptable range.

(5) *“Figure 3 is captioned: Structural characterization of S-closed. But the structure presented is for the “S-closed+Fd” construct.”*

Good catch. We added “Fd” to the title.

Reviewer #3

“Langedijk and colleagues describe the design and biophysical and structural characterization of a number of stabilized prefusion SARS-CoV-2 S trimer variants. They ultimately settled on a variant combining several mutations that is stabilized in the prefusion, 3-down (although not completely) conformation and expresses at higher levels than previous constructs even without a heterologous trimerization domain. Given the extreme impact of SARS-CoV-2 on the world and the potential utility of these new S variants in vaccine development, these results are highly significant and will be of interest to a broad audience. The work is well-laid out and the experiments straightforward and well-designed. The quality of the data is high. I recommend that the article be published in Nat Comms after minor revision.”

Major comments:

“My only major comment is that the text could use some polishing; there are many inconsistencies in formatting, editing, and syntax”

We polished the text and removed formatting and editing inconsistencies we were able to find.

Minor comments:

“Page 5: The authors should clarify that they measure trimer content by analytical SEC. They should further clarify, using a few words in the main text what the nature of the samples were that were loaded onto the column. This information is provided in the materials & methods, but it would help the reader to provide it very briefly in the main text as well.”

We added the underlined parts on page 7, line 128 “in accordance with the results obtained with SEC and AlphaLISA, both in cleared crude supernatants”

“Fig 1A: For all SEC figures, please use Elution volume (mL) instead of Time (min) on the x axis. Also, please specify in the figure caption the column that was used in these experiments.”

We adjusted all SEC figures to show Elution volume (mL) instead of Time (min) on the x axis. We added “on an SRT-10C SEC-500 15 cm column” to the legends of SEC figures.

“Fig S2: The y axis should be changed to indicate that the derivatives of the fluorescent signal are plotted. Please also include the raw fluorescence signal (not the derivative) for completeness.”

We changed the y axis label to “Fluorescent Signal (Derivative, A.U)”. We added figure S2B to show the raw fluorescence signals.

“Page 7 / Fig 1D: Please use a few words to clarify what assay was used to measure antibody binding. Also, while the initial rates of increase of BLI signal may be an appropriate measure of

relative binding, the raw BLI curves must be provided as supplementary material to allow the reader to interpret the data on their own. Also, the y axis in Fig 1D should be relabeled something like V_0 (nm/s)."

We added the underlined text on page 7 line 127: "The variant with K986P showed higher binding of SAD-S35, ACE2 and CR3022 than S-2P, measured with BioLayer Interferometry (Fig. 1D left panel)". Details of the assay can be found in the Methods section.

Full BLI curves are now provided in Figure S5 (former Figure S6).

We adjusted the label of the axis in Fig1D to V_0 and modified the legend accordingly. The same was done in Figure 2D.

"Fig 2A: The authors should provide ACE2 binding data for S-closed. Also, there is no need for scientific notation on the x axis tick labels"

The reviewer has a good point here. Unfortunately, the ACE2 binding in supernatant could not be determined for S-closed. In the process of removing the foldon, also the C-terminal tag was removed from S-closed+Fd. Without this tag we could not measure ACE2 binding to S protein in a comparable manner (AlphaLISA) as for the other variants, because the tag is needed for detection in a two-bead-based AlphaLISA. Therefore, we could not generate comparable data for S-closed.

We changed the x axis tick labels.

"Fig S6: The authors should indicate in the caption that the data pertain to Fig 2D."

In the new expanded legend of Figure S5 (former Figure S6) we added "The curves were used to derive the initial slope V_0 plotted in Figures 1D and 2D".

"Page 10 / Fig 3: The authors should provide a slightly more detailed description of their structure of S-closed. What is the RMSD of their model compared to that of Walls et al.? Are there any regions that deviate significantly? Are there backbone-level changes near the mutations, or only changes in side chain configurations? Also, the Fig 3 caption should state whether the semi-transparent surface representations are actual density or are calculated from the model. These representations should be consistent across all four sub-panels (currently missing from the D614 panel). Also, mutations should be indicated as such using the residue labels (e.g., A942P), while non-mutated residues should be labeled as they currently are (e.g., V860)."

We have calculated the all atom RMSD value with the Walls structure and added it in the text. We have also made a comment about differences between the two structures. Page 10, line 184: "The two structures differ by 2.2 Å all atom RMSD and there are no significant differences in terms of relative position of domains or domain conformation. The RBD is relatively more defined and the NTD less defined compared to the closed trimer described by Walls et al."

The mutations do not significantly affect the backbone conformation when compared to known wild-type structures. We have added in the text (Page 11 line 187): “The stabilizing mutations do not significantly affect backbone conformation of the closed trimer.”

In order to make Figure 3 subpanels more consistent, the transparent surface representation was removed. We believe it made the figure clearer, since adding the surface representation on top of the hydrogen bonds in panel b obfuscated the presented interaction. The subpanels were replotted using the final structure.

Mutated amino-acids are now clearly indicated in Fig 3 a-d panels, as requested.

“Page 12, line 189: Other groups have described this instability as well, and their work should be cited here.”

We added the following references: Wrapp et al 2020, Walls et al 2020, Hsieh et al 2020, Xiong et al 2020, Cai et al 2020, Henderson et 2020, McCallum 2020. The text now says (Page 13, line 201, previously page 12, line 189): “SARS-CoV-2 S protein is unstable^{9,12,23-27}”

Reviewer Comments second round:

Reviewer #1 (Remarks to the Author):

The authors have done a good job in replying to my comments, and I also feel that they addressed reasonably the issues raised by the other reviewers. I find the revised manuscript as acceptable for publication

Reviewer #2 (Remarks to the Author):

The authors have adequately answered all my comments.

With regards to storing spike supernatants at 4 °C, the authors have explained in detail that these proteins were stored for 1-2 days at 4 °C and were assessed similarly. The authors should add some of these details to the methods, where they currently state: "The spun supernatant was subsequently 283 sterile filtered using a 0.22 µm vacuum filter and stored at 4°C until use." which is ambiguous and needs to be better defined especially for proteins that are known to be unstable in cold. "until use" could mean 1 week, 1 month, or any length of time. Based on their responses, I suggest the following wording: "The spun supernatant was subsequently 283 sterile filtered using a 0.22 µm vacuum filter and stored at 4°C until they were purified within 1-2 days of harvest."

Reviewer #3 (Remarks to the Author):

Langedijk et al. addressed all of the reviewers' comments where possible, and the manuscript is, in my opinion, acceptable for publication in Nature Communications.

REVIEWERS' COMMENTS

Reviewer #1 (Remarks to the Author):

The authors have done a good job in replying to my comments, and I also feel that they addressed reasonably the issues raised by the other reviewers. I find the revised manuscript as acceptable for publication

Reviewer #2 (Remarks to the Author):

The authors have adequately answered all my comments.

With regards to storing spike supernatants at 4 °C, the authors have explained in detail that these proteins were stored for 1-2 days at 4 °C and were assessed similarly. The authors should add some of these details to the methods, where they currently state: "The spun supernatant was subsequently 283 sterile filtered using a 0.22 µm vacuum filter and stored at 4°C until use." which is ambiguous and needs to be better defined especially for proteins that are known to be unstable in cold. "until use" could mean 1 week, 1 month, or any length of time. Based on their responses, I suggest the following wording: "The spun supernatant was subsequently 283 sterile filtered using a 0.22 µm vacuum filter and stored at 4°C until they were purified within 1-2 days of harvest."

We have incorporated Reviewer's #2 suggestion in Materials and Methods section.

Reviewer #3 (Remarks to the Author):

Langedijk et al. addressed all of the reviewers' comments where possible, and the manuscript is, in my opinion, acceptable for publication in Nature Communications.